# Supported MOCVD TiO_2_ Thin Films Grown on Modified Stainless Steel Mesh for Sensing Applications

**DOI:** 10.3390/nano13192678

**Published:** 2023-09-29

**Authors:** Naida El Habra, Francesca Visentin, Francesca Russo, Alessandro Galenda, Alessia Famengo, Marzio Rancan, Maria Losurdo, Lidia Armelao

**Affiliations:** 1Institute of Condensed Matter Chemistry and Technologies for Energy (ICMATE), National Research Council (CNR), Corso Stati Uniti 4, 35127 Padova, Italy; francesca.visentin@cnr.it (F.V.); alessandro.galenda@cnr.it (A.G.); alessia.famengo@cnr.it (A.F.); maria.losurdo@cnr.it (M.L.); 2Department of Chemical Sciences, University of Padova, Via Marzolo 1, 35131 Padova, Italy; francescarusso21@hotmail.com (F.R.); lidia.armelao@cnr.it (L.A.); 3Institute of Condensed Matter Chemistry and Technologies for Energy (ICMATE), National Research Council (CNR), c/o Department of Chemical Sciences, University of Padova, Via Marzolo 1, 35131 Padova, Italy; marzio.rancan@cnr.it; 4Department of Chemical Sciences and Materials Technologies (DSCTM), National Research Council (CNR), Piazzale A. Moro 7, 00185 Roma, Italy

**Keywords:** COD sensors, wet acid etching, MOCVD, supported TiO_2_, photocatalysis, increased surface area substrates

## Abstract

Among semiconductor metal oxides, that are an important class of sensing materials, titanium dioxide (TiO_2_) thin films are widely employed as sensors because of their high chemical and mechanical stability in harsh environments, non-toxicity, eco-compatibility, and photocatalytic properties. TiO_2_-based chemical oxygen demand (COD) sensors exploit the photocatalytic properties of TiO_2_ in inducing the oxidation of organic compounds to CO_2_. In this work, we discuss nanostructured TiO_2_ thin films grown via low-pressure metal organic chemical vapor deposition (MOCVD) on metallic AISI 316 mesh. To increase the surface sensing area, different inorganic acid-based chemical etching protocols have been developed, determining the optimal experimental conditions for adequate substrate roughness. Both chemically etched pristine meshes and the MOCVD-coated ones have been studied by scanning electron microscopy (SEM), X-ray diffraction (XRD), energy dispersive X-ray (EDX) microanalysis, and X-ray photoelectron spectroscopy (XPS). We demonstrate that etching by HCl/H_2_SO_4_ at 55 °C provides the most suitable surface morphology. To investigate the behavior of the developed high surface area TiO_2_ thin films as COD sensors, photocatalytic degradation of functional model pollutants based on ISO 10678:2010 has been tested, showing for the best performing acid-etched mesh coated with polycrystalline TiO_2_ an increase of 60% in activity, and degrading 66 µmol of MB per square meter per hour.

## 1. Introduction

The detection, quantification, and monitoring of chemical and biological compounds have become of significant interest in the last decades [1,2]. Sensors can be manufactured with several active materials, depending on the measurement principle exploited or the target compounds to be detected [1,2]. The recent drastic increase in human population and industrialization has made better environmental management and, therefore, the development of more accurate and timely systems for detecting water pollutants necessary, as also stated by UN Sustainable Development Goal 6. Chemical oxygen demand (COD) is an indicator for assessing the concentration of organic contaminants in water resources, directly affecting the health of aquatic systems [1,3,4,5]. Traditional COD techniques require expensive and toxic reagents to induce oxidative degradation, involving complex, lengthy, time-consuming analyses [4,5,6,7]. Lately, the utilization of titanium dioxide (TiO_2_) in the thin film form as a COD sensor has been developing as a promising and innovative approach to rapid, accurate, and real-time monitoring of water quality in industries, urban wastewater treatment facilities, and other environmental situations [1,2,5,6].

Titania-based COD sensors employ photocatalytic advanced oxidation processes (AOPs) that originate when the TiO_2_ semiconductor is irradiated by photons having energy equal to or higher than the band gap [8], promoting electrons (e^−^) to the conduction band, leaving holes (h^+^) in the valence band. These e^−^ and h^+^ can react with electron acceptors and donators molecules, therefore starting redox processes and causing the mineralization of the pollutants [5,7,9]. The COD can be calculated by measuring the number of the photogenerated electrons through the dissolved oxygen determination or by monitoring the change in metal ions’ oxidation states under UV irradiation [2,5]. TiO_2_-based COD sensors are promising devices thanks to titania excellent oxidative capabilities under UV irradiation, high photostability, low toxicity, and cheapness [2,5,7].

To fully exploit the titania-based material as COD sensors, several parameters of the active devices must be optimized. The generally used powder suspensions of the photocatalytic material have a high surface-to-volume ratio [5]; conversely, these systems require a processing step to remove the catalyst from aqueous suspensions and reuse it. Metal-supported photocatalyst consists of the coating of a suitable substrate with a thin layer of photoactive material to be easily recovered and reused [8,10]. Nevertheless, the immobilization of the catalyst inevitably reduces the density of surface-active sites, leading to a loss of photocatalytic activity. The utilization of high surface-area substrates could be a way to overcome this problem, consequently leading to enhanced COD quantification sensitivity and accuracy.

To increase the surface roughness by creating a hierarchical structure of the supporting substrates (generally stainless steel), different methods have been applied, such as sputtering, anodization, lithography, compressed metal powders, electrodeposition, and etching processes [11,12,13,14]. One of the most used techniques for metal substrates is the etching procedure. Plasma-assisted etching processes are dry methods that use plasmas to remove material from the substrate surface. They allow high accuracy and precision in the surface topography, but remove a very small amount of material [15]. Moreover, the size of the object to be treated is generally limited by vacuum chamber size, and in plasmas operating at low pressure, the vacuum system is expensive and requires heavy maintenance. The wet chemical etching technique, on the other hand, uses strong acid or alkaline chemical solutions to remove part of the workpiece by controlling its dissolution [16] and is one of the most attractive micromachining processes to enhance surface roughness [16]. It is low-cost, simple, and suited for large-area production involving bulk metal removal. Wet chemical etching is fast, with high selectivity, and with a controllable etching rate, which depends on the material features (crystallinity, grains size, and defects), the concentration and temperature of the etchants, and the etching time [12,14,16,17,18].

Few studies have focused on the surface modification of stainless steel surfaces through wet chemical etching by using FeCl_3_ solutions (a commonly used etchant in the industry) mixed with acid solutions (such as HCl, H_2_SO_4_, HNO_3_, HF, etc.) [12,13]. Matthews et al. [19] demonstrated that concentrated FeCl_3_ solution mixed with concentrated HNO_3_ and HF causes rapid and uniform corrosion of the stainless steel. Furthermore, Snyder et al. [20] used different compositions of aqueous solutions of HCl, FeCl_3_, and HNO_3_ with wetting agents to increase the corrosion rate and the uniformity of the stainless steel chemical attack.

After the etching step, a suitable deposition technique may be used to grow the (photo)active sensing TiO_2_ material in thin film form on the supporting substrate. Various deposition techniques have been used, such as sol–gel, electrodeposition, direct oxidation, electron beam evaporation, pulsed laser deposition, etc. [2,5,21,22]. Among them, low-pressure metal organic chemical vapor deposition (LP-MOCVD) is widely used for depositing high-quality polycrystalline TiO_2_ thin films, at relatively low temperatures, with controlled deposition rates, good reproducibility, adhesion, and with high conformal properties, also on complex-shaped surfaces [10,23,24]. The characteristics of the coating, such as chemical composition, crystallinity, microstructure, and thickness, can be easily modified simply by working on the deposition parameters [25]. Additionally, the MOCVD process is also prone to large-scale production.

In this work, stainless steel meshes have been used as supporting substrates for TiO_2_-based COD sensor devices. Different inorganic acid-based chemical etching protocols have been first tested to increase the surface area of the pristine supporting mesh. MOCVD TiO_2_ coatings deposited on chemically etched stainless steel meshes have been investigated in relation to the type of etchant solutions. A complete structural, morphological, and compositional characterization has been performed by scanning electron microscopy (SEM), X-ray diffraction, energy dispersive X-ray (EDX) microanalysis, and X-ray photoelectron spectroscopy (XPS).

Moreover, for preliminarily testing the potentiality of the TiO_2_ thin films deposited on functionalized mesh substrates as COD sensors, photocatalytic degradation tests on model pollutants based on ISO 10678:2010 have been reported and discussed. The ISO exploits the photoactivated degradation on methylene blue in an aqueous solution to estimate the material photocatalytic properties. The experiments have been carried out to compare the functional properties of TiO_2_ thin films deposited on functionalized and pristine mesh substrates.

## 2. Materials and Methods

### 2.1. Acid-Based Chemical Etching of Stainless Steel Mesh

The pristine stainless steel mesh (AISI 316, named SS) used as supporting substrate and subsequently subjected to the acid-based chemical etching was purchased from a local provider (Gaudenzi Srl, Albignasego, Padova, Italy). The pristine meshes were made of crossed wires of 28 µm nominal diameter and 27.8 µm nominal hole size. The mesh pieces were square portions of 2 × 2 cm^2^. Before use, the meshes were washed with water and soap (Labkem Cleaner M66), rinsed with deionized water and ethanol, and air dried at room temperature (RT).

Three different acid solutions were used to etch the meshes: HCl (37%, Sigma-Aldric ACS reagent grade), H_2_SO_4_ (95–98%, Fluka^TM^ ACS reagent grade), and a mixture of HCl/H_2_SO_4_. All the experiments were carried out using 50 mL of a 10M acid solution, freshly prepared before each experiment (see Appendix A). The meshes were immersed into the solution under continuous stirring and maintained at different temperatures, from 25 °C (i.e., RT) up to 60 °C, and for different durations (10, 15, and 25 min, see Table 1). After each etching treatment, a cleaning process was applied to remove any acid residue with deionized water and ultrasound (two successive cycles of 10 mins). Finally, the meshes were rinsed with ethanol and air dried at RT. The meshes after the acid treatment are named SS-AT.

The best-performing etching solutions were identified on small pieces of nets (4 cm^2^) and then transferred to a more extended area mesh (18.75 cm^2^) by proportioning the volumes of the 10M acid solutions to the new area value.

### 2.2. Deposition of Nanostructured TiO_2_ Thin Films via MOCVD

TiO_2_ depositions were carried out on extended area meshes of 7.5 × 2.5 cm^2^. TiO_2_ depositions on the bare mesh (sample SS-TiO_2_) and the best acid-etched mash (sample SS-AT-TiO_2_) were carried out using a custom-made cross-flow MOCVD reactor equipped with a hot-wall chamber and operating at low pressure. Titanium tetra-isopropoxide [Ti(O*^i^*Pr)_4_] (TTIP, 97%, with O*^i^*Pr = -CH(CH_3_)_2_, from Sigma-Aldrich) was used as the titanium precursor (without any further purification). TTIP was kept at 50 °C and delivered inside the reaction chamber by a 110 sccm N_2_ carrier flow. The TTIP pipeline between the precursor tank and the reactor was kept at 75 °C to avoid precursor condensation. MOCVD experiments were carried out at a pressure of 0.30 Torr and a temperature of 395 °C. No co-reactant precursors were used. Different deposition times were investigated to obtain the desired film thickness (20, 25, 30, 33, 35, and 40 min).

### 2.3. Microstructural Investigations

SS mesh, SS-AT, SS-TiO_2_, and SS-AT-TiO_2_ were characterized by different techniques to establish their structure–property correlations with the different acid treatments and deposition conditions.

The cross sections and surface morphologies were investigated by a Quanta 200 FEG ESEM Scanning Electron Microscopy ((SEM), FEI, Hillsboro, OR, USA), equipped with a field emission gun, operating in high vacuum conditions and at an accelerating voltage of 20–25 kV, in relation to the observation needs. Energy dispersive X-ray (EDX) measurements were performed for qualitative and semiquantitative element investigation of the samples and were performed by using an EDAX Genesis energy dispersive X-ray spectrometer. All the EDX atomic percentages (at.%) are reported as mean values, deriving from three different measurements conducted on each sample. Each acquisition was detected from 500× area micrographs accumulated for 100 s at 25 kV and a working distance of 10 mm.

The structural characterization of TiO_2_ deposits was performed by X-ray diffraction (XRD) using a Bruker D8 Advance Plus (Bruker, Karlsruhe, Germany) diffractometer, operating in Bragg–Brentano geometry equipped with Cu X-ray radiation (λ = 1.5406 Å, 40 kV, and 40 mA). The phase identification was supported by the Standard 2002 ICDD database files. The crystallite dimensions were calculated employing Scherrer’s equation.

XPS analyses were performed with a PerkinElmer Φ 5600-ci spectrometer using Al Kα radiation (1486.6 eV), according to a previously described setup [26,27]. Samples were mounted on a steel holder and introduced directly into the fast-entry lock system of the XPS analytical chamber. The XPS spectrometer was calibrated by assuming the BE of the Au 4f_7/2_ line at 83.9 eV with respect to the Fermi level. The BE shifts were corrected by assigning to the C 1s peak associated with adventitious hydrocarbons a value of 284.8 eV. The standard deviation for the BEs values was ± 0.2 eV. The sample analysis area was 800 μm in diameter. Survey scans were obtained in the 0–1350 eV range with the following parameters: 187.8 eV pass energy, 0.8 eV step^−1^, and 0.05 s step^−1^. The parameters used for the detailed scans were 23.5 eV pass energy, 0.1 eV step^−1^, and 0.1 s step^−1^. Data analysis involved Shirley-type background subtraction, nonlinear least squares curve fitting adopting Gaussian–Lorentzian peak shapes, and peak area determination by integration. The atomic compositions were evaluated from peak areas of detailed scans using sensitivity factors supplied by PerkinElmer.

### 2.4. Photocatalytic Measurements

The photocatalytic activity of the SS, SS-TiO_2_, and SS-AT-TiO_2_ samples was analyzed with respect to the ISO standard 10678:2010 [28] for methylene blue (MB) removal, also described in previous works [9,10]. Briefly, ISO 10678:2010 controls the determination of photocatalytic properties of surfaces in an aqueous medium through methylene blue degradation. ISO standard suggests communicating the results as specific photoactivity P_MB_ (mol·m^−2^·h^−1^) and photonic efficiency ζ_MB_ (%), calculated by the equations provided in the protocol. The specific photoactivity is defined as the methylene blue amount converted per unit of catalyst surface in the unit of time—the difference in the degradation rate of the solution kept in the dark (reference) and exposed to UV radiation (irradiated). Photonic efficiency represents the amount (in percentage) of useful photons in MB degradation with respect to the total amount of incident photons. It is assumed that one photon induces the bleaching of one MB molecule. UVA Philips PL-S 9W/2P BLB at 370 nm lamps were employed in the present case, with 10 ± 0.5 W/m^2^ measured at the height of the sample. Delta Ohm HD 2302.0 light meter equipped with an LP471PA probe was used for measuring the power density of the lamps. The measurements were performed at 22 °C (using a thermostatic bath) in a beaker filled with 50 mL methylene blue solution (10 μmol/L for test solution and 20 μmol/L for conditioning solution) on samples with a geometrical surface of 4.00 cm^2^. The measurements were carried out following the standard protocol, as well as the data collection and processing, while the MB solution absorbance was measured by a Shimadzu UV-2600 spectrophotometer.

## 3. Results

### 3.1. Increasing the SS Surface Area: Acid-Based Etchings

#### 3.1.1. HCl, H_2_SO_4_, and HCl/H_2_SO_4_ Etchings: Morphological Characterization

SEM surface analyses were carried out on the pristine SS mesh to investigate the pre-exposure to acids structure, as shown in Figure 1. Micrographs highlight a regular mesh of the bare net with a smooth surface; only some groves are evident, deriving from the machining process.

For each acid solution (HCl, H_2_SO_4_, and HCl/H_2_SO_4_), the effect of different temperatures and immersion times were tested. The acid treatment effects were followed through SEM surface analyses by observing the corrosion response on SS boundary grains in relation to the different adopted conditions.

In Figure 2, the HCl, H_2_SO_4_, and HCl/H_2_SO_4_ etched meshes are reported after 25 °C (RT) treatment for 10, 15, and 20 min of immersion. The SS-AT-HCl sample shows irregular boundary grains and demonstrates a slight effect on SS net corrosion after 15 and 20 min. At RT for HCl solution, the corrosive effect induced by the prolongation of the acid treatment times starts to be slightly evident after 15 min of immersion (Figure 2a–c). Cl^–^ ions bring metals in solution from the SS mesh in the forms of metals’ chlorides, specifically iron chlorides [16]. The etching is not observed after H_2_SO_4_ treatment; the surface of the mesh remains smooth and without relevant morphological variations (Figure 2d–f). At RT, the HCl/H_2_SO_4_ solution does not significantly affect the mesh morphology, except for some grain boundary irregularities, analogously to what was observed for the single HCl treatment (Figure 2g–i).

HCl and HCl/H_2_SO_4_ etching procedures significantly affect the morphology of the sample treated at 40 °C (Figure 3a–c,g–i). Indeed, SEM images show a selective intergranular erosion with grooves between each boundary grain, wider than those observed at 25 °C (Figure 2a–c,g–i). This behavior is enhanced by prolonging the immersion time (from 15 to 20 min), which induces more marked and deep grooves after 20 min, especially for the HCl solution. However, the surface of the grain does not appear to be affected by the etching and still seems flat and smooth. On the other hand, the treatment with single H_2_SO_4_ (Figure 3d–f) at the same temperature does not evidence any morphological surface variation, maintaining a flat and smooth appearance without evident grain boundaries.

As widely reported in the literature [16,29], to increase the etching rates, the temperature of acid solutions is raised to 60 °C (Figure 4). A remarkable etching process is observed using both HCl and HCl/H_2_SO_4_, also evidenced by the vigorous reaction of the net just after its immersion in the solutions and the change in the solution color from colorless to dark green. However, after these treatments, the meshes immediately appear to lose their mechanical properties, becoming more flaccid and thinned. In Figure 4, the SS-AT meshes, treated in the three different acid baths at 60 °C for 20 min (Figure 4b–f), are compared with the untreated SS one (Figure 4a). The negative effect of the acid exposition at high temperatures is evidenced by the reduction in the wires’ thickness, up to about 10% compared with the original dimension. The enhancement of the bath temperature induces a high increase in the pristine SS surface roughness, with pronounced intragrains’ asperities and, consequently, the complete loss of the grains’ flatness, previously observed at lower etching temperature (Figure 4d,f). According to Cremaldi et al. [13], the combination of high temperatures and long etching times induces a high surface roughness deriving from the corrosion of the more resistant metals (such as chromium) in the SS mesh, leading to the formation of protrusions, more evident as the immersion time increases. As expected, the treatment at high temperatures and for a prolonged time (20 min) with sulfuric acid reduces the effect on SS-AT mesh when compared with HCl and HCl/H_2_SO_4_ etching solutions. In Figure 4e, an extended pitting is shown, characterized by small holes of about 300–500 nm. Moreover, the grain boundaries’ erosion and the mechanical properties loss are not observed under these conditions.

The acid treatments carried out at 40 °C are too mild for the purposes of this work, while those at 60 °C are detrimental to the mechanical properties of the treated meshes. For this reason, intermediate temperatures of 50 and 55 °C are tested with HCl and HCl/ H_2_SO_4_ solutions. The H_2_SO_4_ solution, having demonstrated to be inappropriate to produce a sufficient chemical etching even at 60 °C, is instead abandoned.

SEM micrographs of the meshes treated with HCl and HCl/H_2_SO_4_ solutions after the treatment at 50 °C and 55 °C are shown in Figure 5 for different immersion times (10, 15, and 20 min). At the same temperature, increasing the time of the acid treatment induces an enhancement of the etching effect. This behavior is particularly effective for the meshes treated at 55 °C for 20 min (for both the HCl and HCl/H_2_SO_4_ solutions), where the disappearance of the flat grain regions can be observed with a concomitant increase in the densities of asperities/protrusions and the intergrains’ distances. Moreover, unlike the etching at 60 °C, the mesh maintains its mechanical properties, and the wire dimensions are not significantly changed.

For HCl/H_2_SO_4_ etching solutions, the two different temperatures showed different effects. At 50 °C, the increase in the immersion time results in a higher definition of grain boundaries, even if the etching effect is not so evident as previously observed for the only HCl treatment and for the same time of immersion (Figure 5a–c). This different performance may be ascribed to a competition between the etching activity of the two acids in the solution. It is reasonable to assume that the HCl corrosive effect is initially inhibited by the presence of H_2_SO_4_. At 55 °C and just after 15 min of treatment, the HCl/H_2_SO_4_ solution shows marked intergranular corrosion characterized by high asperities and voids density (Figure 5k), with a progressive increase in surface roughness as the time of immersion extends (Figure 5l), comparable to the results obtained with the single HCl solution (Figure 5h,i).

XRD patterns acquired after the different acid etching treatments at 55 °C for 15 min (SS-AT) together with the SS-pristine mesh are reported, for the sake of brevity, in Appendix A. All the analyzed samples show austenitic steel (cubic γ-phase) characteristic of AISI 316. The three primary reflections at 2θ = 43.66°, 50.84°, and 74.67° derive from (111), (200), and (220) planes, respectively (ICDD 00-031-0619). Moreover, for the SS-pristine mesh, the (110) peak of the Fe-rich martensitic tetragonal phase (α’-phase, ICDD 00-044-1292) at 44.68° can be easily distinguished from the main (111) peak of the cubic γ-phase, while it progressively becomes less detectable after H_2_SO_4_, HCl, and HCl/H_2_SO_4_ etching treatments, respectively. No preferential orientation has been detected after the acid treatments.

#### 3.1.2. HCl, H_2_SO_4_, and HCl/H_2_SO_4_ Etchings: EDX Characterization

The temperature influence of the chemical etching solution is among the most important factors affecting the metal removal kinetics. The literature reports that an increase in the temperature of the acid bath leads to greater corrosive efficiency and, consequently, an increase in the etching speed [29].

To semiquantitatively determine the chemical composition of the mesh surfaces as a function of temperature and immersion times, EDX measurements are carried out. On SS meshes, EDX analyses identified elements such as Fe, Cr, Mo, Ni, O, and C. In this work, for brevity, the behavior of the main mesh constituents, Cr and Fe, are monitored.

To determine the effect of the type of acid used in relation to the temperature of the acid bath, all the analyzed meshes are treated for the same time (15 min), as reported in Figure 6. In the 25–60 °C temperature range, the leaching effect of the acid bath is maximum at T = 60 °C for SS-AT-HCl (Figure 6a), with both Fe atomic percentages (at.%) and Cr at.% markedly lower than the values measured at RT, 40 °C, and 55 °C. However, an overall decreasing trend can be observed for Fe at.% as a function of the temperature increasing, while for Cr at.%, the EDX values are comparable in the RT–55 °C temperature range. The Fe and Cr at.% important reduction at 60 °C is consistent with the poor mechanical properties of the nets observed by SEM analyses. Moreover, preferential corrosion of Fe with respect to Cr is found, indicating a major resistance of chromium to the HCl attack.

A different behavior than HCl is observed after 15 min of H_2_SO_4_ etching (Figure 6b): Fe at.% and Cr at.% are similar in the whole temperature range considered (RT–60 °C), and they do not vary with respect to the SS-pristine mesh. These data can be reasonably interpreted considering the complexity of corrosion processes in concentrated H_2_SO_4_ solutions. Rouzmeh et al. [30] found that H_2_SO_4_ at pH 1.5 causes the formation of a thick layer of iron oxo-hydroxide species at the surface of mild steel, originating from the deposition of corrosion products because the solution behaves as a strong acid able to protonate the iron oxides present on the surface [31]. Similarly, Cr at.% is expected to be comparable to the SS-pristine mesh because of the oxidative passivation occurring at high H_2_SO_4_ concentrations, leading to the deposition of Cr oxide/oxo-hydroxide layers, as observed by Stypula et al. [32] for Cr corrosion in H_2_SO_4_. Furthermore, in the review published by Panossian et al. [33], it was observed that at H_2_SO_4_ concentrations greater than 9M, the corrosion rate of several types of steel tends to be low. All these observations agree with the constant trend observed for Cr at.% and Fe at. %. This behavior justifies what was evidenced in previously shown SEM images (Figure 2d–f, Figure 3d–f and Figure 4e); the surface corrosion of the SS-AT-H_2_SO_4_ mesh is significantly lower than that of SS-AT-HCl and SS-AT-HCl/H_2_SO_4_ at the same temperature and for the same treatment time.

Finally, in Figure 6c, a different trend is observed for the Fe and Cr at.% after the etching with the HCl/H_2_SO_4_ solution. In this case, competitive phenomena can be assumed between the processes induced by the HCl (corrosion) and the H_2_SO_4_ (passivation). A minimum value at about 40 °C was detected for both Fe at.% and Cr at.%. EDX data support a synergistic effect of the two acids at 40 °C; the etching is sensibly improved with respect to the single acid attack, as shown by comparing Figure 6a–c at 40 °C. However, at higher temperatures up to 55 °C, the metals’ atomic percentages start to increase, and this can be correlated to the deposition of a passivating layer of corrosion products. At 60 °C, the depletion of both Fe and Cr is observed, and this can be due to the concomitant dissolution of the passivating layer and corrosion of the mesh surface. Another important feature is that at 60 °C, the atomic percentages of iron and chromium are comparable to those detected after HCl-only treatment, indicating the prevailing corrosive effect of HCl.

Figure 7 shows the trends of Fe and Cr atomic percentages as a function of immersion time (10, 15, and 20 min) for the three acid solutions at 40 °C and 60 °C. The behavior is consistent with the temperature-dependent trends observed in Figure 6. In general, the etching in HCl (Figure 7a,b) shows an increase in metals’ etching as a function of immersion times, and this effect is more evident at high temperatures (at 60 °C, Fe and Cr at.% curves highlight a higher slope than the ones obtained at 40 °C).

H_2_SO_4_ etching solution (Figure 7c,d) exhibits almost constant Fe and Cr at.% values at 40 °C, whereas at 60 °C, Fe and Cr at.% increase at low immersion times. Differently, after 15 min, a trend inversion is observed. This tendency may be attributed to the combined effect of the high temperature and treatment time, which starts to induce incipient surface corrosion of the pristine SS, reducing the oxidizing properties of the sulfuric acid and favoring the release of metal sulfate into the etching solution. However, as previously observed (Figure 4), the etching for a long time at high temperatures provokes the loss of the meshes’ mechanical properties. For this reason, the treatment with H_2_SO_4_ was discarded.

The treatment at 40 °C with HCl/H_2_SO_4_ (Figure 7e) shows a probable prevailing corrosive effect induced by the HCl acid at up to 15 min of treatment, with a reduction in metals’ at.%. However, after this period, the competitive passivating effect of H_2_SO_4_ starts the surface oxidation of the SS mesh. At 60 °C (Figure 7f), the HCl corrosion is the predominant contributor, also reinforced by the H_2_SO_4_ presence (see micrographs reported in Figure 4).

#### 3.1.3. HCl, H_2_SO_4_, and HCl/H_2_SO_4_ Etchings: XPS Characterization

For the completeness of analyses and to better understand the surface processes induced by the different etching solutions, XPS analyses (Figure 8) are carried out on SS pristine net and etched SS meshes, after immersion in the three acid solutions, for 15 min at 55 °C. As shown in the extended surveys (Figure 8a), the main elements identified are C, O, Cr, and Fe. Generally, no elements such as Cl and S (resulting from the acid baths) are evident. High-resolution measurements were collected for the photoemission peaks C 1s, O 1s, Cr 2p, and Fe 2p. Appendix A reports the atomic percentages of these elements in the four analyzed samples.

Figure 8b shows the O 1s region for the sample treated with HCl. The samples treated with the other acid solutions give similar spectra. The detailed analysis of the O 1s region consistently reveals a contribution from oxygen atoms in the oxide environment (530 eV ca.) and a component with higher binding energy ascribable to both hydroxides/oxo-hydroxides, which are not discernible (at about 531.5 eV). Fitting of O 1s peaks shows that acid treatment with HCl does not induce any variation, with respect to the SS pristine mesh, in the ratio between oxide and hydroxide contribution, while H_2_SO_4_ and the mixture HCl/H_2_SO_4_ induce an increase in the amount of OH groups (Table 2).

Considering the pristine SS net, the Fe 2p_3/2_ BE (711.1 eV), the position of its satellite peak (719 eV ca.), and the whole spectrum shape suggest that iron is Fe(III), probably as Fe_2_O_3_ or FeOOH [29,30,34,35,36] (Figure 8c). All the acid-treated samples also show a contribution at lower BE (706.9 eV) due to Fe(0) (Figure 8d). The appearance of metallic iron after the acid treatments could be attributed to a preferential etching of the surface species (iron oxides/hydroxides), releasing the underlying metallic Fe, a constituent element of the net steel itself. In contrast, the BE of the Cr 2p_3/2_ region (Figure 8e) does not show any variation among the different samples. Cr 2p_3/2_ is centered at 576.8 eV, and such value is typical of Cr(III) and compatible with both chromium oxide (Cr_2_O_3_) and chromium hydroxide [35], as confirmed by Rokosz et al. [29].

The experimental studies, reported in this Section 3.1, demonstrate that, except for the H_2_SO_4_ solution, the etching temperature is one of the most important parameters that affect the kinetics of the metals’ removal. Specifically, a higher etching effect is observed increasing the temperature of the acid solution. In our case, the highest temperature choice was principally limited by the SS mesh, showing poor mechanical properties and reduced wires’ thickness at 60 °C. Furthermore, higher working temperatures could also not be convenient because, as reported by Cakir et al. [16], most etching equipment only allows using a maximum etching temperature range of 50–55 °C. Moreover, it is also detected that the etching time is of fundamental importance for the preparation of different surface-quality functionalized mesh. It is shown that, for HCl and HCl/H_2_SO_4_ solutions, increasing the immersion time in the acid solution increases the surface roughness. This behavior could not be applied to the H_2_SO_4_ solution (as shown in Figure 7), where H_2_SO_4_ passivating properties enhance the chromium fraction on the SS surface, increasing the corrosion resistance of the steel [37]. A higher Cr fraction is detected in XPS analyses (as shown in Appendix A) and confirmed elsewhere [29,37]. Rokosz et al. [29] support the claim that the passivation process is related to an enhancement of Cr fraction in the surface passive film due to a mechanism that involves the selective dissolution of predominantly iron. Rouzmeh et al. [30,31] have compared the efficacy of three different acid solutions, HCl, H_2_SO_4_, and H_3_PO_4,_ and observed that sulfuric acid was the less corrosive of the three acids tested, whereas HCl was the highest corrosive one. They found that the hydroxide species was the highest for the sulfuric acids-treated samples and the lowest for the hydrochloric acid-treated ones, affirming that H_2_SO_4_ and HCl solutions provide the highest and the lowest hydroxide/oxide ratios, respectively. Considering our results reported in Table 2, the oxide/hydroxide ratio exhibits the same behavior found by Roumezh et al. [31], confirming an oxidizing effect induced by the presence of H_2_SO_4_ in the etching solution. However, in the presence of only H_2_SO_4_, its oxidizing effect prevails in the corrosive one, and the meshes show a flat and smooth surface (Figure 2 and Figure 3). In the mixture HCl/H_2_SO_4,_ the corrosive effect of HCl is predominant, leading to a corrosion effect even at room temperature, but obtaining better results by increasing the temperature and immersion time (Figure 2, Figure 3 and Figure 5). Indeed, the appearance of grain boundaries is attributable to the effect of hydrochloric acid, which acts as a source of chlorine, dissolving the surface metals into the solution in the form of metal chlorides [16], in particular Fe chlorides. The probable formation of FeCl_3_ in solution (a chemical agent widely used for the chemical etching of stainless steels) further increases the dissolving capacity of the metal [12,16].

Considering the aim of the present work to increase the surface area of SS meshes to accommodate supported TiO_2_ thin films with higher efficiency and potential as COD sensors (here tested in terms of photocatalytic activity, as reported in the following section), the best results are obtained for the HCl and HCl/H_2_SO_4_ etching solutions operating at 55 °C, for 15 and 20 min of immersion. The choice of the appropriate etching bath is made in view of industrial applications, also considering the costs. In Appendix A, the physicochemical properties, the costs of the single acids (based on Sigma-Aldrich quotations), and the respective solutions are reported. The HCl solution is the most expensive, whereas the H_2_SO_4_ is the least expensive. However, sulfuric acid alone does not lead to an effective increase in the surface area of the net, whereas the HCl/H_2_SO_4_ solution is cheaper (by about 28%) than the hydrochloric acid one, producing surfaces of good quality. Definitively, the immersion time selected is 15 min because of the comparable results obtained at 20 min (see micrographs reported in Figure 5k,l).

### 3.2. TiO_2_ Deposition via MOCVD on SS and SS-AT Meshes

The high surface area meshes (SS-AT) used as supporting substrates for the deposition of the nanostructured TiO_2_ thin films were prepared by acid treatment of the SS nets with the HCl/H_2_SO_4_ mixture and maintained for 15 min at 55 °C, as described in Section 3. Unlike the optimization of the etching procedure, the MOCVD deposition was performed on larger nets, with a total area of 18.75 cm^2^. Therefore, the amounts of acid solutions (optimized in Section 3 for nets of 4 cm^2^) have been proportioned to the dimensions of the networks used (98.4 mL of conc. HCl and 65.6 mL of conc. H_2_SO_4_, mixed with 70.3 mL of deionized water).

Figure 9 reports the SEM micrographs of MOCVD titania thin films deposited with different deposition times (20, 25, 30, 33, 35, and 40 min). For all the deposition times, the TiO_2_ thin films are highly conformal, adhering to the SS-AT substrates and perfectly reproducing their morphological characteristics. The MOCVD process allows the coating of both the upper and lower face of the nets, also covering the inner walls of the meshes.

From low magnifications micrographs (Figure 9a,c,e,g,i,k), it can be seen that increasing the deposition times from shorter (20 and 25 min) to longer periods (30, 33, 35, and 40 min), results in a progressive filling of the intergranular spaces (previously generated during the acid etching) of the networks, with a maximum effect seen on the 40 min sample (Figure 9k), showing the reduction in the surface area for the nets coated with the thicker films. High magnification SEM images (Figure 9b,d,f,h,j,l) show the distinctive morphology of titanium dioxide thin films, characterized by a compact structure of densely packed crystals with a prismatic faceted aspect, dimensionally variable in relation to the deposition time (the grain sizes tend to increase when the film thickness increases). For 20 min of deposition (Figure 9b), the morphology of the film appears different, with a petal-like structure, which is characteristic of low-thickness titania film [10]. Finally, Figure 9m,n show uncoated and TiO_2_-coated nets. As reported in the literature [38], the surface of thin TiO_2_ films (about a few hundred nanometers) acquires a characteristic color due to light interference phenomena taking place at the metal–oxide–air interfaces, and that color depends on the titania film thickness. In our TiO_2_-coated mesh (Figure 9n), the presence of only green and pink colors, together with the absence of interference fringes, indicate a rather regular thickness along the whole length of the coated substrate.

To evaluate the thickness of the titania films on the meshes as a function of the deposition times, SEM cross section analyses were performed. In particular, the cross sections are achieved in the titania-coated pristine SS meshes; the net morphology developed after acid etching made the determination of the thickness via SEM not trivial. Indeed, the optimal film-to-substrate adhesion due to the synergic contributions of mechanical interlocking (related to the increased substrate surface roughness) and of the suitable surface chemistry (the highly hydroxylated surface, see XPS analyses in Section 3.1.3) favors the surface TTIP adsorption. The subsequent triggering of TiO_2_ film deposition [39] made it impossible to analyze the film cross section by cutting the sample.

Figure 10 reports the cross sections of different SS-TiO_2_ samples as a function of the deposition times, from 20 to 40 min. From a morphological point of view, the growth observed in the cross section does not appear properly columnar (i.e., with evident independent vertical columns) but more like a dense interpenetration of the prismatic-like structures, starting at the bottom as thin faceted columns and becoming wider on the top. This behavior appears to be very similar to what is reported in the literature [10,40]. The switch between the thinner and wider column growth seems to occur at a thickness higher than 400 nm. Indeed, such structural variations are not found in thinner films (Figure 10a,b). Table 3 summarizes the growth time of the films, the average thickness of the SS-TiO_2_ deposits obtained from the cross section SEM analyses, and the corresponding growth rates. It can be observed that the growth rates are almost constant, confirming the optimal reproducibility of the MOCVD process performed in our experimental conditions.

XRD analysis of the TiO_2_ films (not shown for brevity) grown on the SS meshes shows that all the films are composed of polycrystalline anatase phase (ICDD n. 01-073-1764) without any evident preferential orientation. In all the samples, the most intense reflection is (101) centered at 2θ = 25.62°, as reported in the 01-073-1764 ICDD. In addition, other characteristic reflections of the anatase phase are detected at 2θ = 48.36°, 54.37°, and 55.38°, deriving from the (200), (105), and (211) planes, respectively. No rutile phase is identified (see Appendix A). The size of the crystallites is calculated using the Debey–-Sherrer’s Equation for the most intense reflections (101) and (200), and the final crystallite size is calculated as the average of the two values. The medium crystallite sizes are 26 nm, 40 nm, 52 nm, 56 nm, 61 nm, and 68 nm, corresponding to 20, 25, 30, 33, 35, and 40 min of deposition, respectively. These values indicate that, by maintaining constant all the deposition parameters, the crystallite size increases linearly as the growth time increases.

As described in ref. [10], the photocatalytic performance depends on several factors, such as the catalyst crystallinity, crystallographic phase, the specific surface area, the electron–hole pairs recombination rate, and the film thickness. In general, the photocatalytic performance is improved by a high crystallinity degree because defects are recombination centers for electron–hole pairs [41]. Furthermore, it is extensively reported that the anatase phase provides better results than the rutile phase. Additionally, anatase/rutile heterojunction could improve performance further, while other crystallographic phases of TiO_2_, such as brookite, are not very active [10]. In the present work, the adopted CVD procedure allows a pure anatase phase with no rutile presence. Considering the optimal film thickness, it must be taken into account that most of the incoming light is absorbed in the first tenths of nanometers of the film. For thinner films, the increase in the film thickness involves the increase in the absorbed incident light and then an increase in the number of photoelectron–hole pair generation. However, this trend is not linear, and for thicker films, a plateau is finally reached. Previous studies [10] have shown that the photocatalytic performances of TiO_2_ films deposited on pristine SS meshes are stable after a critical minimum thickness of about 250–300 nm. The best photocatalytic performance, with comparable efficiency, is obtained from samples with thickness in the 300–940 nm range, with a medium thickness of about 550 nm. In our case, to simultaneously ensure high surface area values and greater photocatalytic activity, the deposit must be thin enough to preserve the morphology of the underlying substrate. It was before shown that a longer period of deposition induces a progressive filling of the intergranular network spaces, indicating a progressive reduction in the net surface area. Considering all these factors, for the successive photocatalytic tests, 33 min (thickness of 545 ± 19 nm) was chosen as the optimal deposition time.

### 3.3. Photocatalytic Activity

The efficiency and performances of COD sensors are strongly influenced by different parameters connected to COD device fabrications, optimization, and design (i.e., the surface area of photocatalytic material with respect to the volume of the polluted water, the microfluidic design, the flow rate of the sample injection, the COD calibration stage, etc.) [5,6]. Therefore, simpler photocatalytic degradation tests on functional model pollutants are performed to compare the photocatalytic performances of TiO_2_ films deposited on both pristine and acid-etched mesh substrates. Specifically, the surface photoactivity was tested through the ISO 10678:2010 standard for photocatalytic-specific activity as a smart tool for highlighting the general surface response. Specifically, in this case, it is worth underlining that the photoactivity improvements with respect to the bare SS meshes should be given by the specific surface area enhancement (thanks to the acid treatment) and the TiO_2_ thin film deposition. To provide a complete data set, the photoactivity, and the photonic efficiency were determined for the bare supporting mesh (SS, i.e., without any kind of treatment), for the TiO_2_ thin film on the bare supporting mesh (SS-TiO_2_), and for the TiO_2_, after 20 min (330 ± 5 nm) and 33 min (545 ± 19 nm) of deposition, on the acid treated supporting mesh (SS-AT-TiO_2_). Each measurement was repeated four times, and the average values of the specific photocatalytic activity, photonic efficiency, and standard deviations were given.

Figure 11a resumes the specific photoactivity outcomes, while in Appendix A, the normalized trend (at [MB]_t*i*_/[MB]_t0_) versus time for the pristine stainless steel mesh, stainless steel mesh with TiO_2_ deposition (33 min), and acid-treated mesh with TiO_2_ deposition (33 min) are reported. The results display the negligible activity of the bare mesh (SS), while the TiO_2_ presence clearly shows significant performance improvements. In detail, the SS-TiO_2_ 33 min removes about 40 µmol of MB per square meter per hour, while the TiO_2_ on the acid-treated mesh reaches 26 and 66 µmol/m^2^·h of MB for the SS-AT-TiO_2_ 20 min and SS-AT-TiO_2_ 33 min, respectively, with more than 60% activity increase for the best-performing sample (SS-AT-TiO_2_ 33 min).

Figure 11b, on the other hand, shows the photonic efficiency for the investigated samples. The observed trend retraces the specific photoactivity, with SS-AT-TiO_2_ at 33 min of deposition able to exploit almost 0.06% of the incident photons, compared with less than 0.04% of the SS-TiO_2_ at the same deposition time. In contrast, the SS-AT-TiO_2_ at 20 min demonstrates about 0.02% of photonic efficiency. As expected, the thinner sample (SS-AT-TiO_2_ 20 min) shows limited photoactivity due to the imperfect thickness–crystallinity morphology combination. The thickness of this sample (about 330 nm) seems to be very close to the critical threshold thickness for reaching the maximum performance. On the other hand, the crystallinity appears low, and the morphology of the sample shows small petal-like structures (about 90–110 nm), which recall samples #60 and #58 in Galenda et al. [10] with scarce activity.

The results obtained for the best-performing sample (SS-AT-TiO_2_ 33 min) agree with those reported for similar systems in previous works. Indeed, the specific activity and photonic efficiency of the SS-TiO_2_ sample are similar to the outcomes reported in ref. [10,42,43]. Since the TiO_2_ film thickness of the investigated samples is comparable (the same experimental parameters were used to coat SS and SS-AT meshes), the high surface area acid-treated supporting mesh appears to be efficiently improved to enhance the photoactivity performance by acting on the specific surface area. The SEM characterization confirms the surface increase as a result of the acid treatment, as shown and discussed in Figure 2, Figure 3 and Figure 5.

## 4. Conclusions

In conclusion, to fully exploit the TiO_2_ performance as a chemical oxygen demand sensor, nanostructured TiO_2_ thin films have been grown via low-pressure metal organic chemical vapor deposition on metallic AISI 316 mesh, whose surface area has been increased by testing different inorganic acid-based chemical etching protocols. The best etching results have been obtained using a single-acid solution of HCl or a mixed-acid solution of HCl/H_2_SO_4_, operating at 55 °C for 15 min. At higher etching temperatures, the pristine mechanical properties of the mesh are lost, with a reduction in the wires’ thickness. The final choice of the HCl/H_2_SO_4_ solution as the best-performing etching system is supported by economic aspects.

To preliminary investigate the detection properties of the developed high surface area-supported TiO_2_ thin films as COD sensors, photocatalytic degradation tests on functional model pollutants, based on ISO 10678:2010, have been carried out. Bare SS mesh showed insignificant photocatalytic activity, SS-TiO_2_ mesh at 33 min was able to remove about 40 µmol of MB per square meter per hour, while the best performing SS-AT-TiO_2_ mesh at 33 min degraded 66 µmol of MB per square meter per hour, with more than 60% of activity gained. The etching treatment, therefore, can be employed as an efficient and relatively economical method to increase the surface area of the metallic supporting substrate and, consequently, the photocatalytic activity of the TiO_2_-coated systems.

These preliminary outcomes demonstrate several promising advantages of high surface area TiO_2_-supported COD sensors towards a more efficient, cost-effective, and environmentally sustainable method of water quality assessment. Our future work will follow several directions:(i)Considering the scalability of the CVD technology for the growth of the TiO_2_ films, we will investigate the scalability of the etching uniformity and the possibility of enlarging the size of the AISI 316 mesh to target industrial-scale sensing application;(ii)Considering that the efficiency of the etching process could also depend on the mesh composition and impurity, it is worth optimizing and developing protocols for other low-cost meshes;(iii)As many performance parameters depend on the structure of the TiO_2_, specifically sensitivity, lifetime, and the working range, a further direction of our study will be unveiling the effect of the mesh microstructure on the time response of the TiO_2_ COD sensor;(iv)In order to move towards wastewater applications, the selectivity and sensitivity to specific organic pollutants, such as repeatable detection of blue-methylene as a probe of main organic contaminants of water, will be investigated.

## Figures and Tables

**Figure 1 nanomaterials-13-02678-f001:**
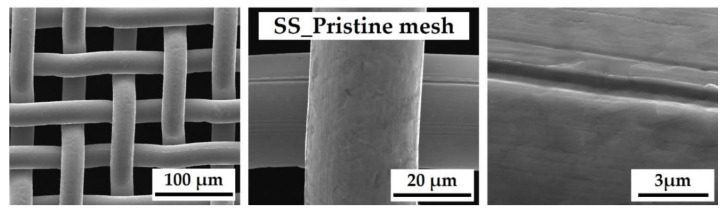
SEM micrographs, at different magnifications (from left to right: 1000×, 5000×, and 30,000×), of pristine SS mesh before the acid etching.

**Figure 2 nanomaterials-13-02678-f002:**
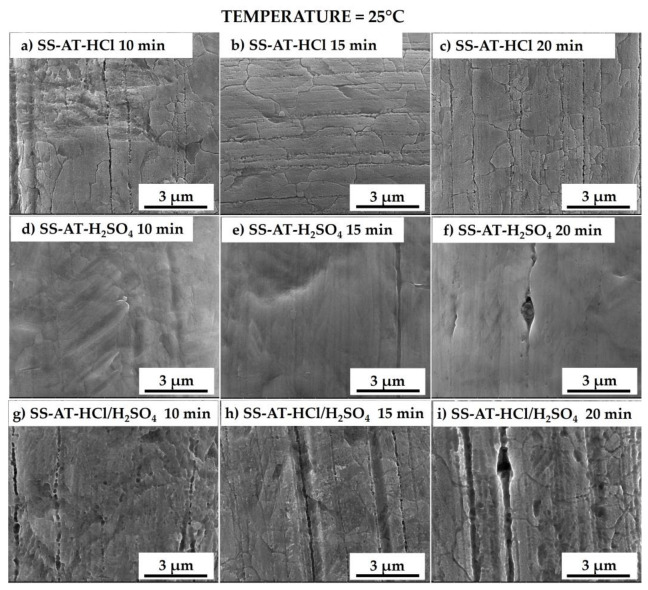
SEM micrographs of the meshes etched at 25 °C with the HCl solution 10M for 10, 15, and 20 min (**a**–**c**), with the H_2_SO_4_ solution 10M for 10, 15, and 20 min (**d**–**f**), and with the HCl/H_2_SO_4_ solution 10M for 10, 15, and 20 min (**g**–**i**). Low-magnification micrographs are reported in Appendix A.

**Figure 3 nanomaterials-13-02678-f003:**
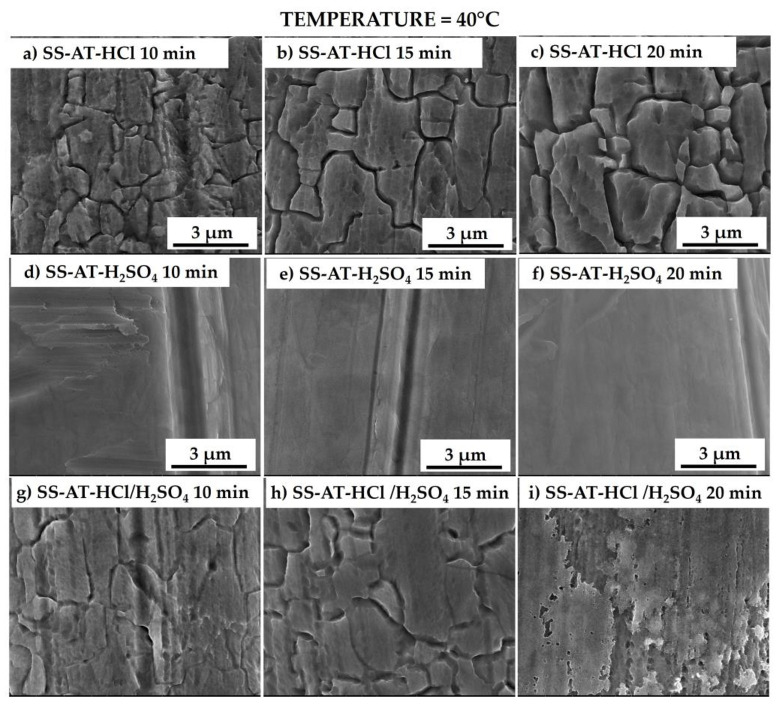
SEM micrographs of the meshes etched at 40 °C with the HCl solution 10M for 10, 15, and 20 min (**a**–**c**), with the H_2_SO_4_ solution 10M for 10, 15, and 20 min (**d**–**f**), and with the HCl/H_2_SO_4_ solution 10M for 10, 15, and 20 min (**g**–**i**). Low-magnification micrographs are reported in Appendix A.

**Figure 4 nanomaterials-13-02678-f004:**
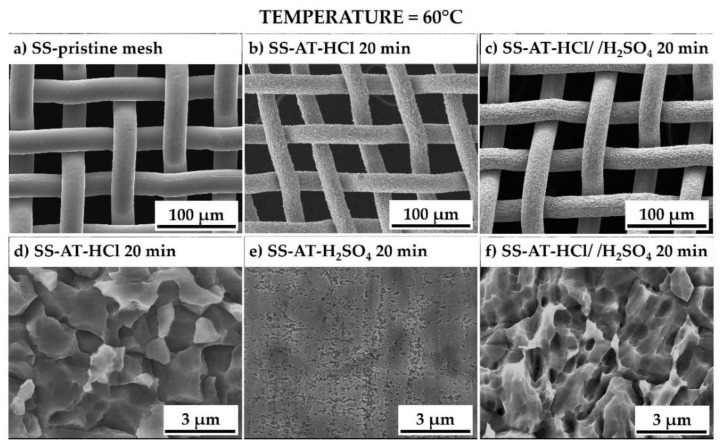
SEM micrographs, at different magnifications, of (**a**) SS-pristine mesh, (**b**,**d**) SS-AT mesh etched by HCl, (**c**,**f**) SS-AT mesh etched by HCl/H_2_SO_4_, and (**e**) SS-AT mesh etched by H_2_SO_4_ solution, at 60 °C and for 20 min of immersion.

**Figure 5 nanomaterials-13-02678-f005:**
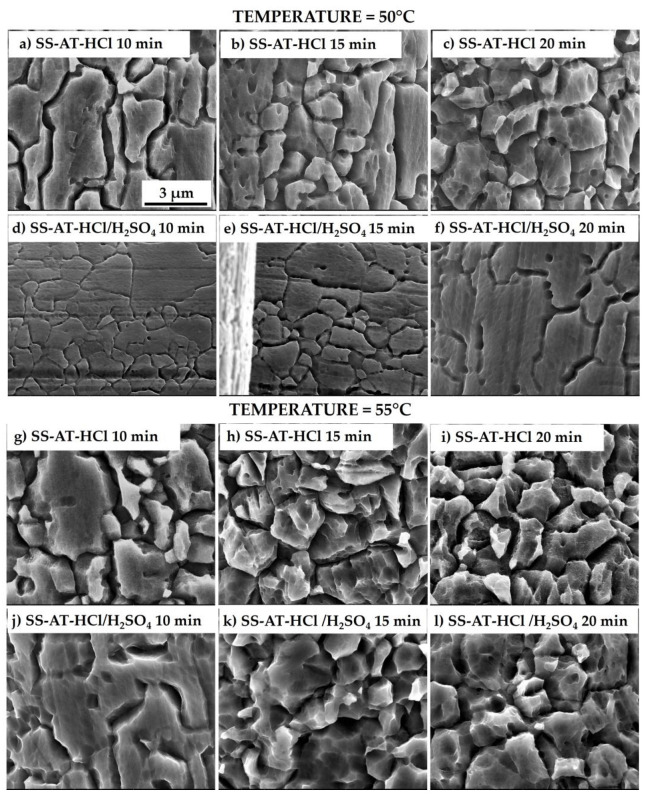
SEM micrographs of SS-AT meshes etched by HCl solution at 50 °C for (**a**) 10 min, (**b**) 15 min, and (**c**) 20 min, and by HCl/H_2_SO_4_ solution at 50 °C for (**d**) 10 min, (**e**) 15 min, and (**f**) 20 min. SEM micrographs of SS-AT meshes etched by HCl solution at 55 °C for (**g**) 10 min, (**h**) 15 min, and (**i**) 20 min, and by HCl/H_2_SO_4_ solution at 55 °C for (**j**) 10 min, (**k**) 15 min, and (**l**) 20 min.

**Figure 6 nanomaterials-13-02678-f006:**
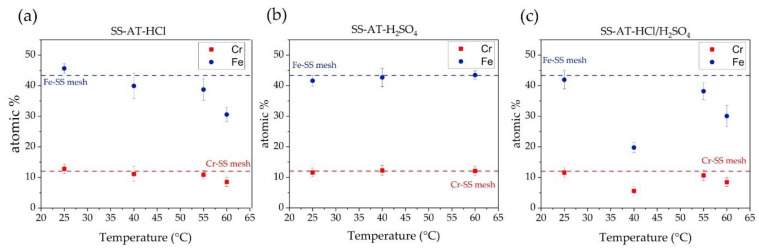
Trends of EDX atomic percentages of Fe and Cr elements in SS-AT meshes as a function of the etching temperature after chemical etching for 15 min using (**a**) HCl, (**b**) H_2_SO_4_, and (**c**) HCl/H_2_SO_4_ acid solutions. The dashed lines correspond to the at.% of Fe (blue line) and Cr (red line) in the SS-pristine mesh.

**Figure 7 nanomaterials-13-02678-f007:**
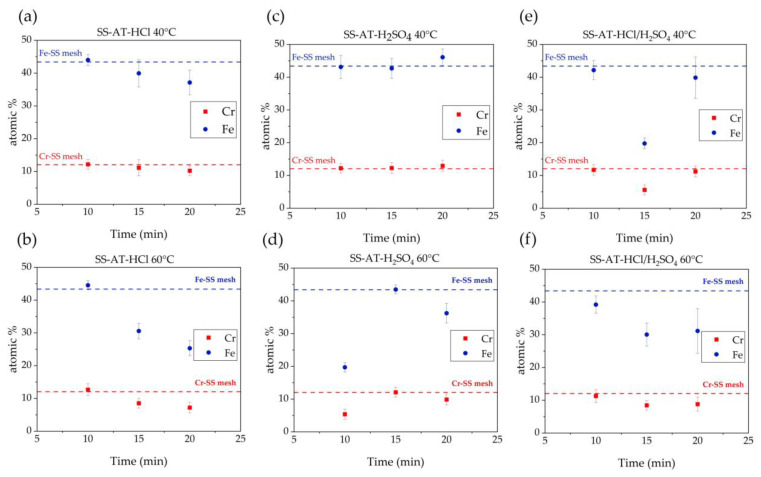
Trends of EDX atomic percentages of Fe and Cr elements in SS-AT meshes as a function of the etching immersion times, after chemical etching at 40 °C and 60 °C, for (**a**,**b**) HCl, (**c**,**d**) H_2_SO_4_, and (**e**,**f**) HCl/H_2_SO_4_ acid solutions. The dashed lines correspond to the at% of Fe (blue line) and Cr (red line) in the pristine SS-mesh.

**Figure 8 nanomaterials-13-02678-f008:**
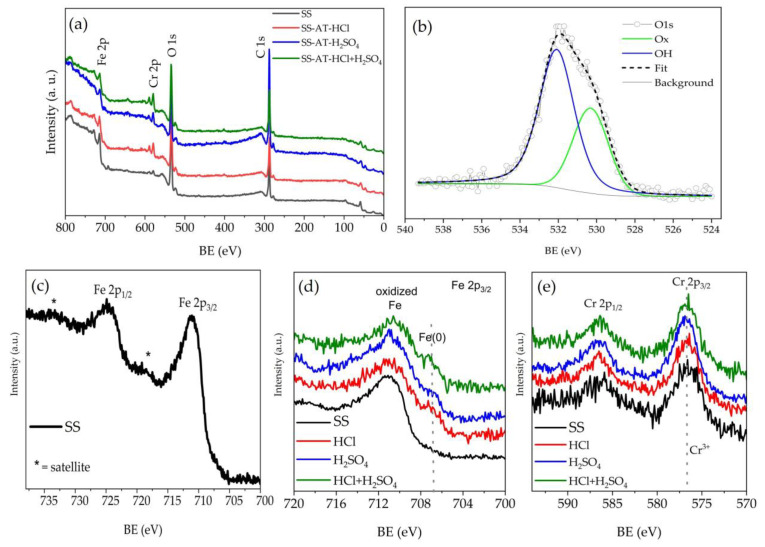
(**a**) XPS survey spectra. (**b**) O1s photoemission peaks and fitting for SS-AT treated with HCl etching solution. (**c**) Fe 2p region of the pristine SS net. (**d**) Fe 2p_3/2_ region for the pristine SS and SS-AT nets. (**e**) Cr 2p region for the pristine SS and SS-AT nets.

**Figure 9 nanomaterials-13-02678-f009:**
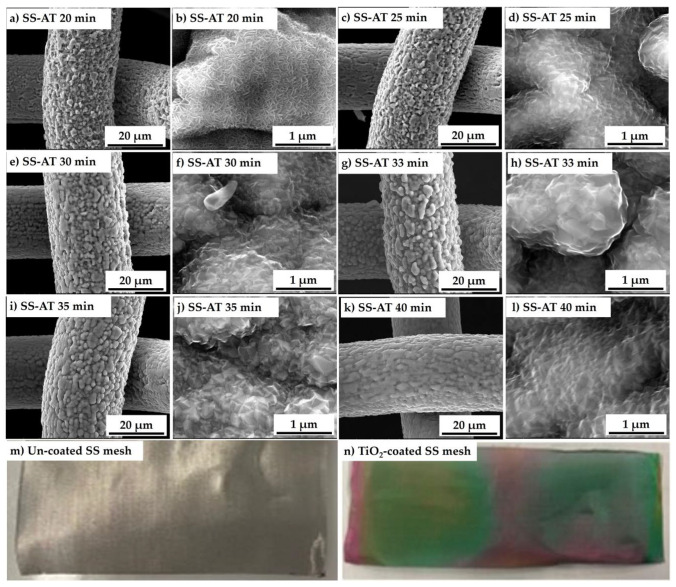
SEM images at different magnifications (5000× and 100,000×) of TiO_2_ deposits on acidified networks (SS-AT-TiO_2_) at different deposition times of (**a**,**b**) 20 min, (**c**,**d**) 25 min, (**e**,**f**) 30 min, (**g**,**h**) 33 min, (**i**,**j**) 35 min, and (**k**,**l**) 40 min. Photos of uncoated SS (**m**) and SS titania-coated (**n**) meshes.

**Figure 10 nanomaterials-13-02678-f010:**
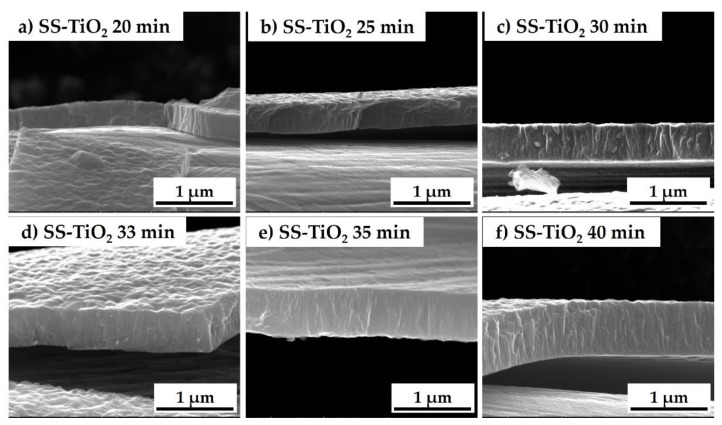
Cross section SEM images of titania films deposited on SS mesh for different deposition times: (**a**) 20 min, (**b**) 25 min, (**c**) 30 min, (**d**) 33 min, (**e**) 35 min, and (**f**) 40 min.

**Figure 11 nanomaterials-13-02678-f011:**
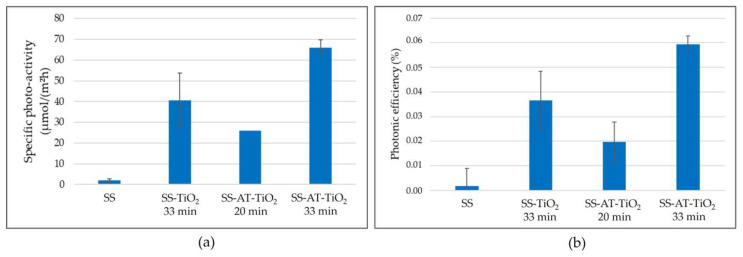
(**a**) Specific photoactivity for the SS, SS-TiO_2_, and SS-AT-TiO_2_ meshes; (**b**) Photonic efficiency for the SS, SS-TiO_2_, and SS-AT-TiO_2_ meshes.

**Table 1 nanomaterials-13-02678-t001:** Working temperatures and durations employed for the etching treatments of stainless steel meshes.

Acid Solution (10M)	Temperature (°C)	Treatment Duration (Min)
HCl	25–40–50–55–60	10–15–20
H_2_SO_4_	25–40–60	10–15–20
HCl/H_2_SO_4_	25–40–50–55–60	10–15–20

**Table 2 nanomaterials-13-02678-t002:** Oxide and hydroxide percentages resulting from the fitting of O 1s photoemission peaks.

	Oxide %	Hydroxide %	Oxide/Hydroxide Ratio
SS	42.2	57.8	0.7
SS-AT-HCl	41.9	58.1	0.7
SS-AT-H_2_SO_4_	34.0	66.0	0.5
SS-AT-HCl/H_2_SO_4_	33.7	66.3	0.5

**Table 3 nanomaterials-13-02678-t003:** Thicknesses of the TiO_2_ MOCVD thin films deposited on SS-pristine mesh as a function of the deposition times. Mean growth rates are also calculated.

**Deposition Time (Min)**	**Thickness (nm)**	**Growth Rate (nm/Min)**
20	330 ± 5	16.5 ± 0.2
25	390 ± 11	15.6 ± 0.4
30	481 ± 17	16.0 ± 0.6
33	545 ± 19	16.5 ± 0.6
35	631 ± 18	18.0 ± 0.5
40	713 ± 21	17.8 ± 0.5

## Data Availability

Not applicable.

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
