# Peer review of "Supported MOCVD TiO2 Thin Films Grown on Modified Stainless Steel Mesh for Sensing Applications"

_nanomaterials, 2023, doi:10.3390/nano13192678_

Round 1

Reviewer 1 Report

Please, find my review in the attached file

Author Response

Please see the atttachment.

Reviewer 2 Report

The authors present a manuscript that deals with MOCVD TiO2 thin films grown on modified stainless-steel mesh for sensing applications. I think the topic is quite interesting and the study could be published although I have a few comments

Towards the end of the abstract the tests of the constructed sensors are briefly mentioned.

" To preliminary investigate the detection properties of the developed high surface area TiO2 thin films as COD sensors, photocatalytic degradation tests on functional model pollutants based on ISO 10678:2010 have been reported and discussed."

However, as this is an abstract there should be a proper summary of the catalytic/sensing results 

Introduction

I think this needs clarification, to be rewritten more scientifically " TiO2-based COD sensors are highly stimulating thanks to titania's excellent oxidative capabilities under UV irradiation and high photostability, coupled with its non-toxic, inexpensive, and eco-compatible nature [2,5,7]."

General comment that the introduction is quite long and gives a step by step development of this type of sensor. Although it is well referenced it could perhaps be shortened slightly to provide more of an overview of the state of the art in this type of material/sensor.

End of introduction could be improved - this sentence has some English issues but also it is replicated from the current abstract. Again could give more specific detaisl of the aims/objectives of the study and the test procedures

"Moreover, for preliminarily testing the potentiality of the TiO2 thin films deposited on functionalized mesh substrates as COD sensors, photocatalytic degradation tests on model pollutants based on ISO 10678:2010 have been reported and discussed, with the aim to compare the functional properties of TiO2 thin films deposited on functionalized and pristine mesh substrates."

Experimental section

Needs more details or better description i.e. detergent (xxx)

"Before their use, the mesh was washed with water and soap,"

Could Table 1 be moved to supplementary material as it just details production of diluted acid? It may be more important to know the purity and source of the chemicals used.

Line 156 - requires details - "operating at low pressure" may be detailed further down in the text?

Results - not sure why figure 1 is separate from the acid etched samples as this does not aid comparison?

Not sure we need so many micrographs in the main text of the paper - possibly move some to supplementary materials?

A general comment abouit the results - the treatment of the etching process is quite exhaustive. I understand this is an important aspect but I feel the functionality of the material is equally so. Therefore, the authors should try to balance these aspects in the manuscript. I feel that the results need to be condensed and only the major results presented within the text of the paper for clarity - currently there is too much information and this is detracting from the main findings of the work in my opinion.

It is not that clear what figure 6 shows, some display trends but some profiles are difficult to explain?

Not sure why table 5 needs to be in the main text of the paper?

Conclusions: Need to have a focus on future developments required for the current system in addition to a more general focus on future applications which is included.

In places the use of English detracts from the quality of the work. 

Reviewer 3 Report

This manuscript deals mainly with the structural and sensing properties of supported TiO2 thin films grown on modified stainless-steel mesh.  Both chemically etched pristine meshes and  MOCVD-coated ones were studied by SEM, XRD, EDX and XPS. And photocatalytic  degradation tests on functional model   pollutants based on ISO 10678:2010 were perforned, which revealed that SS-AT-TiO2 mesh can degrade 66 mu mol of MB per square meter per hour. This article performed interesting study on sensing properties of supported TiO2 thin films grown on modified stainless-steel mesh, and the results would be helpful for the exploration of highly efficient COD sensors. This article was generally clearly  written and well organized. So, this paper can be accepted for publication. In addition, some minor revisions may be considered by the authors. 1) In Tables 2 and 5, some subscripts were not correctly displayed. Please check them. 2) At the end of Sect. 3, a brief comparison in sensing properties between present samples and some popular products would be helpful. 3) Is it possible that blunting effect of iron in concentrated sulfuric acid may influence the structural and sensing properties of the samples? Please clarify. 

Reviewer 4 Report

In this work, nanostructured TiO2 thin films have been grown via Low-Pressure Metal Organic Chemical Vapor Deposition (MOCVD) on metallic AISI 316 mesh as supporting substrates. I think there are many problems with this article, and it is recommended to be revised after publication. I have these questions.
1 The authors need to provide the spectra XRD after corrosion.

2. Articles need to be polished and the abstract needs to be rewritten.

3. In order to more define the structural samples, the authors need to provide TEM test results.

4. Authors need to add performance tables comparable to other TiO2 literature.

5. The authors stated that "SS-TiO2 removes about 40 micromolar MB per square meter per hour, while the TiO2 acid-treated mesh reaches MB of 66 μ mol / m2 • h, increasing activity by more than 60%". The authors should explain why the performance improves.

Moderate editing of English language required

Round 2

Reviewer 1 Report

The article was carefully revised and can be published in the present form

Only minor corrections are needed.

Reviewer 2 Report

The authors addressed most of my points. The current manuscript is improved and I feel could be published in the present form if the other reviewers/editors agree.

English is improved from the first version

Reviewer 4 Report

The author addressed the reviewer's comments properly, I recommend its publication.

Minor editing of English language required.